# Dynamic Inverse Relationship Between Cell-Free DNA and Anti-dsDNA Antibodies in Experimental SLE Highlights the Potential for Targeted Immunomodulatory Therapy

**DOI:** 10.3390/pathophysiology32030048

**Published:** 2025-09-16

**Authors:** Mark M. Melamud, Evgeny A. Ermakov, Anna S. Tolmacheva, Georgy A. Nevinsky, Valentina N. Buneva

**Affiliations:** 1Institute of Chemical Biology and Fundamental Medicine, Siberian Branch of the Russian Academy of Sciences, 630090 Novosibirsk, Russia; marken94@mail.ru (M.M.M.); evgeny_ermakov@mail.ru (E.A.E.); tolmacheva.anna0301@gmail.com (A.S.T.); nevinsky@niboch.nsc.ru (G.A.N.); 2Department of Natural Sciences, Novosibirsk State University, 630090 Novosibirsk, Russia

**Keywords:** systemic lupus erythematosus, cell-free DNA, anti-DNA antibodies, pristane, SLE mouse model

## Abstract

**Background/Objectives**: The pathognomonic feature of systemic lupus erythematosus (SLE) is the formation of antibodies to double-stranded DNA (anti-dsDNA Abs). Cell-free DNA (cfDNA) has been suggested as one of the antigens for the generation of anti-dsDNA Abs, but the temporal changes in these biomarkers are not clear. In this study, the association of dynamic changes in total cfDNA and anti-dsDNA Abs levels in blood plasma during disease progression in a murine model of pristane-induced SLE was examined. **Methods**: The experimental group consisted of 12 BALB/c pristane-immunized mice; the control group included 8 PBS-treated mice. Blood samples were collected six times during the 38-week study (2 weeks before and 8, 14, 22, 28, and 36 weeks after immunization). Total cfDNA and anti-dsDNA Abs levels were determined at each time point. **Results**: Pristane-immunized mice showed a significant increase in the concentration of anti-dsDNA Abs. A 14-week delay in the formation of anti-dsDNA Abs was observed after an increase in the concentration of cfDNA in the experimental and control groups. Anti-dsDNA Abs and total cfDNA levels did not correlate at specific time points, but the change in cfDNA concentration from week 14 to week 28 was inversely correlated with the change in the anti-dsDNA Abs level over the same time period (R = −0.71, *p* = 0.009), i.e., the more the anti-dsDNA Abs level increased, the more the cfDNA concentration decreased. A direct correlation was shown between the increase in body weight of pristane-immunized mice and the increase in total cfDNA concentration in the blood from week 0 to week 14 (R = 0.6, *p* = 0.04). **Conclusions**: These findings demonstrate the dynamic nature of cfDNA and anti-dsDNA Abs levels and reciprocal dynamics of these markers in a pristane-induced mouse model of SLE.

## 1. Introduction

Systemic lupus erythematosus (SLE) is a severe rheumatic disease that dramatically affects the quality of life of patients. The prevalence of SLE ranges from 3 to 517 cases per 100,000 population, depending on the region [1]. A systematic review by Baker and Pope found that approximately 34% of SLE patients have work disability [2]. SLE is also associated with a high economic burden. In 2006, Huscher et al. found that the total annual cost of treatment and monitoring for a patient with SLE in Germany was more than 14,000 euros per year [3]. The high economic cost of treatment and the social consequences make it necessary to study the mechanisms of SLE pathogenesis in order to develop new therapeutic drugs.

The main pathognomonic sign of SLE is an increase in the concentration of anti-double-stranded DNA antibodies (anti-dsDNA Abs) in a patient’s blood. Anti-dsDNA Abs play an important role in SLE diagnosis and treatment [4]. At present, the mechanism of anti-dsDNA Abs formation is not fully understood [5]. However, there is some evidence to suggest that cell-free DNA (cfDNA) may play a role in the generation of these antibodies [6]. In addition, the formation of complexes of DNA (including cfDNA) with anti-dsDNA Abs is associated with the pathogenesis of lupus nephritis [6]. CfDNA molecules are considered alarmins because they stimulate the pro-inflammatory immune response [7]. Alarmins are molecules released from dying cells or secreted during sublethal stress. Alarmins, also known as DAMPs (damage-associated molecular patterns), initiate an immune response at the site of damage [8]. There is currently evidence of an inverse correlation between the concentration of cfDNA in the blood of patients with SLE and the level of anti-dsDNA Abs [9] or the absence of any link between these markers measured at any one time point [10]. However, studies on the association between dynamic changes in cfDNA concentration and anti-dsDNA Abs level during the development of SLE have not yet been performed.

The half-life of cfDNA ranges from 15 min to 2 h [11,12,13] and according to more precise data is 24.2 min [14]. DNA is known to be more stable in complexes with proteins such as histones, high-mobility group protein B1 and others. Animal studies have shown that immunization of mice with a DNA-histone complex leads to the formation of Abs with DNA-hydrolyzing activity [15]. In addition, antibodies with high DNA-hydrolyzing activity have been detected in patients with SLE [16,17,18,19]. Thus, stable DNA-histone complexes have greater immunogenicity due to an increased half-life and lead to the formation of a wide repertoire of autoantibodies, including cross-reactive ones [19,20].

In this research, a pristane-induced model of SLE was used. This is one of the most popular induced models of the disease, which has been applied for many years in the study of SLE pathogenesis and treatment [21]. The pristane-induced lupus model is often contrasted with genetic models such as MRL/lpr mice. The pristine-induced model offers several key advantages [21]: it is easy to establish and maintain in a laboratory setting; pristane reliably induces lupus-like manifestations, including anti-dsDNA Abs production and kidney damage; immunized mice develop disease in a predictable timeframe (whereas the time between immunization and manifestation varies significantly in genetic models of SLE [21]), which is crucial for experimental planning; the development of lupus-like symptoms in this model is highly reproducible. The manifestation of lupus-like symptoms, especially anti-dsDNA Abs at predictable time frames with high reproducibility, makes this model particularly valuable when working with small cohorts of animals. These features make the pristane-induced model particularly valuable for mechanistic and therapeutic investigations in SLE. However, changes in cfDNA and anti-dsDNA Abs concentrations in the dynamics of pathology development have not been analyzed in this or other murine models.

The aim of this study was to investigate the association of dynamic changes in cfDNA and anti-dsDNA Abs levels during disease progression in a mouse model of pristane-induced SLE.

## 2. Materials and Methods

### 2.1. Animals and Immunization

The study was performed on 20 female BALB/c mice, 8 weeks old. The mice were obtained from the State Research Centre of Virology and Bacteriology Vector (Novosibirsk, Russia). The study was conducted in accordance with the Declaration of Helsinki and approved by the Local Ethics Committee of the Institute of Chemical Biology and Fundamental Medicine (ICBFM) SB RAS (Protocol No. 3 of 19 June 2023). Throughout the study, the mice were kept in a fully equipped vivarium for laboratory animals at the ICBFM SB RAS (Novosibirsk, Russia). Before the study, all mice underwent a two-week quarantine.

To determine the required sample size, a power analysis was performed. Since no literature data on cfDNA concentrations in pristane-immunized mice were found, human data from SLE patients were used as reference. In most studies of cfDNA concentration in patients with SLE, it was approximately 4-fold (or even more) higher than in healthy donors [9,22,23,24]. Based on published studies, it was assumed that diseased animals would show a 4-fold increase in biomarker levels compared to controls. The standard deviation was estimated as 50% of the mean value in each group, corresponding to a standardized effect size of Cohen’s d ≈ 1.41. With 80% power and a two-tailed significance level of α = 0.05, the calculation indicated that 7–9 animals per group would be required.

The mice were randomly divided into two groups: experimental and control. The experimental group included 12 animals that were immunized with 2,6,10,14-tetramethylpentadecane (pristane) (Sigma-Aldrich, St. Louis, MO, USA) to create a pristane-induced SLE model. A single intraperitoneal injection of Pristane in a volume of 500 μL was performed as in other similar experiments [25]. The control group included 8 of the same animals, which were immunized with a PBS solution (500 μL, intraperitoneally, once). Rapid weight loss of more than 20% during the observation week was considered a humane endpoint requiring euthanasia. However, no animal met this criterion during the experiment.

The experimental design is shown in Figure 1. Blood volume of 200 μL was collected through the retroorbital sinus into capillary blood collection tubes Microvette 200 EDTA K3E (Sarstedt AG & Co, Nümbrecht, Germany). All manipulations were performed by an experienced researcher to minimize stress of the animal. Lidocaine was used as a topical ophthalmic anesthetic agent. Blood plasma was obtained by centrifugation (2000× *g*, 15 min).

### 2.2. Total cfDNA Isolation and Concentration Determination

Total cfDNA was isolated from blood plasma using the blood DNA isolation kit D-Blood-250 (Biolabmix, Novosibirsk, Russia) according to the manufacturer’s instructions. CfDNA concentration analysis was performed using the Qubit dsDNA High Sensitivity Assay Kit on a Qubit 4 fluorimeter (Thermo Fisher Scientific, Waltham, MA, USA).

### 2.3. Concentration of Anti-dsDNA Abs Determination

The concentration of anti-dsDNA Abs was determined by enzyme-linked immunosorbent assay (ELISA) using the Mouse Anti-dsDNA antibody ELISA Kit (Cat. #EM1471, FineTest, Wuhan, China). Optical density was measured using a Multiskan™ FC Microplate Photometer (Thermo Fisher Scientific, Waltham, MA, USA) at a wavelength of 450 nm according to the manufacturer’s recommendations.

### 2.4. Statistical Analysis

Statistical analysis and visualization of the obtained data were carried out in OriginPro 2021 (OriginLab Corporation, Northampton, MA, USA). The type of data distribution in the samples was evaluated using the Shapiro–Wilk test. The Mann–Whitney test was used to compare the values of unrelated samples at a single time point. The Wilcoxon test was used to analyze related samples or the significance of differences within the same group at different time points. Correlations were determined using Spearman’s R correlation test.

## 3. Results

### 3.1. Visual Inspection and Weight Measurement

The mice in the experimental group rapidly developed SLE-like symptoms. Five weeks after the immunization, two of the mice in the experimental group developed alopecia areata. Twenty-eight weeks after the immunization, the mice in the experimental group began to lose their hair coat. These changes in mice from the experimental group are shown in Figure 2a,b.

In addition, a difference in body weight was found between mice from the experimental and control groups during the study (Figure 3). Pristane-immunized mice gained weight faster than the control animals. Starting at 8 weeks after immunization, mice in the experimental group were statistically significantly heavier than control animals (*p* < 0.01), except at 24 and 26 weeks after immunization. They were found to have a marked accumulation in adipose tissue in the abdominal region (Figure 2b).

### 3.2. Dynamic Changes in Plasma cfDNA and Anti-dsDNA Abs Levels

During the first 8 weeks after immunization, no difference in plasma cfDNA concentration was observed between the experimental and control groups (Figure 4a). In Pristane-immunized mice, plasma cfDNA level increased dramatically from 8 to 14 weeks after immunization. Then, there was an equally sharp decline to the initial values. Unexpectedly, plasma cfDNA concentration also increased in the control group from 8 to 22 weeks after immunization. After that, there was also a decline to baseline values. The difference in total cfDNA concentration in plasma between groups was not statistically different at any time point.

In mice immunized with pristane, the concentration of anti-dsDNA Abs began to increase almost immediately, and by week 8 it was already statistically significantly higher than that of the healthy control (*p* = 0.00025) (Figure 4b). After that, from week 8 to week 22 after immunization, the concentration of anti-dsDNA Abs in the plasma of pristane-immunized mice slightly decreased and reached a plateau. By week 22 after immunization, the statistical significance of differences between the groups disappeared (*p* = 0.07). From week 22 after immunization, a sharp increase in the concentration of anti-dsDNA Abs in the blood plasma of mice from both the experimental and control groups began. The anti-dsDNA Abs level in the experimental group reached its peak at week 28 after immunization. At this point, it again became statistically significantly higher than that of the control group (*p* = 0.00045). Then, the anti-dsDNA Abs concentration in plasma of mice from the experimental group began to decrease. In control mice, the concentration of anti-dsDNA Abs was the highest at 36 weeks after immunization. At this point, the statistically significant difference between the groups disappeared for the second time (*p* = 0.06). Detailed changes in cfDNA and anti-dsDNA Abs concentrations in each group between the closest time points are summarized in Appendix A. Thus, initially cfDNA levels increased, and only after 14 weeks did anti-dsDNA Abs levels increase significantly.

### 3.3. Correlations

No statistically significant correlations were found between cfDNA concentration and anti-dsDNA Abs level at any of the time points. However, as shown in Figure 4a,b, the peak of anti-dsDNA Abs was shifted forward by fourteen weeks compared to the peak of cfDNA concentration. It was observed in both the experimental group (weeks 28 and 14) and the control group weeks 36 and 22). Remarkably, a statistically significant inverse correlation was found between the absolute change (delta) in plasma anti-dsDNA Abs and cfDNA concentrations in pristane-immunized mice from weeks 14 to 28 (Figure 5a). No such correlation was found for the control group, but as noted above, there was a 14-week difference in peaks of cfDNA and anti-dsDNA Abs concentration observed. Thus, cfDNA and anti-dsDNA Abs levels were not correlated at individual time points, but changes in the levels of these indices were inversely correlated. In other words, the more the anti-dsDNA Abs level increased, the more the cfDNA concentration decreased.

In mice of the experimental group, a statistically significant direct correlation was found between the increase in body weight and the absolute change (delta) in cfDNA concentration from week 0 to week 14 after immunization (Figure 5b). In the first 14 weeks, mice from the experimental group exhibited the most pronounced increase in weight (including due to abdominal fat tissue) and the concentration of cfDNA in plasma.

## 4. Discussion

The clinical course of the SLE-like condition in mice from the experimental group (Figure 2 and Figure 3) is consistent with the currently available literature. Immunization of mice with pristane has previously been shown to cause alopecia and hair loss [21]. Skin lesions are also common in patients with SLE. Reidin et al. found that alopecia areata is associated with SLE [26]. We found no studies on weight changes in mice immunized with pristane. However, in studies of SLE-prone B6.MRL/lpr mice, there was no significant increase in body weight compared to control C57BL/6 mice [27]. It is possible that pristane, as a toxic external antigen, causes increased stress in mice compared to the genetic mice model. This stress leads to excess food intake and, therefore, weight gain.

There is current evidence that the concentration of cfDNA in peripheral blood increases in SLE patients [9,22,23]. However, the pathogenetic link between cfDNA and anti-dsDNA Abs in SLE has not been fully established [6]. As previously mentioned, there is some published evidence of an inverse correlation between cfDNA and anti-dsDNA Abs at a single time point [9]. In this study, the maximum concentration of cfDNA in the plasma of experimental group mice was detected on the 14th week after immunization (Figure 4a). After that, the cfDNA declined to the initial values. Statistical processing of the data revealed no correlation between the concentrations of cfDNA and anti-dsDNA Abs at each time point, but a correlation between the dynamics of changes in these parameters was found (Figure 5a). Therefore, the more the anti-dsDNA Abs concentration increased in the plasma of experimental group mice from week 14 to week 28 of the study, the more the cfDNA concentration decreased. The inverse cfDNA/anti-dsDNA Abs correlation may be related to antibody-mediated cfDNA clearance and thereby reduced cfDNA half-life. Such dynamic correlations between the parameters studied are more convincing than associations at specific time points because antibody levels reflect antigen concentrations in the past rather than in the present. However, the time lag turned out to be longer than initially assumed. Since similar experiments have not been performed previously, it was hypothesized that the time interval between the increase in cfDNA and anti-dsDNA Abs concentration in blood would be the classic 1–2 weeks, which is typical of both mice [28] and humans [29]. The response time was shown to be about 14 weeks in both groups of mice. This was much longer than expected. We hypothesize that the 14-week delay is associated with slow accumulation of immune-stimulatory complexes, impaired T cell responses or involvement of T cell-independent antibody production mechanisms, and the activity of DNA clearance mechanisms including blood DNases that fragment DNA reducing its antigenic potential. The lag phenomenon in antibody formation is possibly related to differences in the type of antigen presentation compared to other SLE models. Finally, fluctuations of anti-dsDNA Abs may be related to a limitation of the sampling intervals.

The maximum concentration of cfDNA in the blood plasma of mice in the control group was found after 22 weeks of the study; after that it returned to the initial values (Figure 4a). Although the cause of this increase in the control group remains unknown, it helped to confirm a 14-week time lag in the formation of anti-dsDNA Abs, as the maximum concentration of Abs in the plasma of the control mice occurred at week 36 after immunization (Figure 4b). Based on a single study with a small number of animals, it is difficult to explain why the concentration of cfDNA in the plasma of mice in the control group increased by week 22 of the study. Mice were randomly assigned to each of the two groups at the beginning of the study. The homogeneity of the two groups is further supported by the fact that they initially had nearly identical plasma cfDNA concentrations (34 and 21 ng/mL in the pristane-induced and control groups, respectively) (Figure 4a). One possible reason for the increase in cfDNA concentration could be stress [30]. Another potential explanation for the rise in cfDNA levels is age. Human studies have shown that older individuals have higher cfDNA concentrations than younger ones [31]. For mice, this hypothesis is indirectly supported by the observation that anti-dsDNA Abs levels increase with age. More detailed information on this will be provided later in the discussion of changes in anti-dsDNA Abs concentration. Thus, stress, age, and other factors can lead to fluctuations in cfDNA levels. Unlike the experimental group, no statistically significant inverse correlation was found in the control group between the change in the concentration of cfDNA and anti-dsDNA Abs in the plasma of mice. However, as can be seen in Figure 4, the visual pattern of changes in the graphs in the experimental and control groups is very similar. Thus, dynamic changes in the concentration of cfDNA and anti-dsDNA Abs had an inverse relationship in a mouse model of SLE. Consequently, analyzing these indicators at a single time point may not be representative. Therefore, future studies should consider changes in these parameters at least at two time points to examine the relationship of cfDNA and anti-dsDNA Abs levels in mouse models of SLE and in humans.

The pristane-induced model of SLE is characterized by the elevation of anti-dsDNA Abs levels in the blood of mice [21,32]. This study showed that the concentration of anti-dsDNA Abs in the experimental group becomes statistically significantly higher by the 8th week of the study, but a rapid increase occurs in the period between week 22 and week 28 (Figure 4b). This observation is consistent with the findings of He et al. who showed a significant increase in anti-dsDNA Abs concentration in the blood of mice immunized with pristane by week 20 of the study [32]. Another study by Liu et al. also showed an increase in the concentration of anti-dsDNA Abs in the blood of mice immunized with pristane [33]. In this work, the increase in concentration started between months 2 and 3 (week 8 and 12) and a sharp increase occurred around month 5 of the study (week 20) (Figure 4b). Based on the data of this study and the results of other works [32,33], it can be concluded that a dramatic increase in the titer of anti-dsDNA Abs in the blood of mice immunized with pristane occurs from 20 to 25 weeks after immunization.

The observed increase in the concentration of anti-dsDNA Abs in the blood of BALB/c mice of the control group with age is quite difficult to compare with literature data, since the observation period in the above studies is limited to 20 [32] and 24 [33] weeks after immunization. In the present study, the main increase in anti-dsDNA Abs concentration occurred from 22 to 36 weeks (Figure 4b). However, in the study by Liu et al., there was a small increase in the concentration of anti-dsDNA Abs in the control mice between 20 and 24 weeks, but the study was stopped after that [33]. However, other mouse strains have shown an increase in anti-dsDNA Abs level with age [34,35,36], although the observation period was less than 36 weeks. In addition, there is evidence that the prevalence of anti-dsDNA Abs increases with age in healthy humans [36,37]. Therefore, through a long-term experimental design, this study showed an increase in anti-dsDNA Abs formation with age in BALB/c mice that are not susceptible to spontaneous development of autoimmunity. Overall, these results complement existing evidence of increasing levels of autoreactive Abs with age.

The correlation between weight gain in the experimental mice and an increase in the concentration of cfDNA in the blood plasma (Figure 5b) may confirm the link between the excessive formation of adipose tissue and increased inflammatory processes. It was shown that the increase in weight of the mice in the experimental group was caused, among other things, by an increase in the amount of adipose tissue in the abdominal region (Figure 2b). The control mice, which also gained weight due to their maturation, did not show such correlation, since they did not form excess deposits of adipose tissue. At the moment, an association between excess weight and inflammatory markers is identified [38]. However, weight gain may be an indicator of stress or nonspecific immune activation. The direct correlation between weight change and change in cfDNA concentration found in this study may be associated with the mechanism of neutrophil death known as netosis [39]. Neutrophils are an important source of cfDNA released during netosis as part of neutrophil extracellular traps. Numerous studies indicate activation of netosis, including through adipokines and other mediators, in patients living with obesity [40,41]. Therefore, an increased netosis caused by obesity may be one of the reasons for the correlation found. In addition, obesity is known to be a factor that aggravates the course of SLE [42]. Although no association between overweight and increased antibody levels to dsDNA has been shown in the literature, an indirect effect may occur through increased cfDNA formation. The direct correlation of cfDNA levels with weight gain (Figure 5b) found in this study is intriguing and warrants further investigation.

Several limitations should be taken into account in analyzing and evaluating the obtained results. Firstly, although pristane immunization of mice is one of the most common models of SLE, it is still difficult to extrapolate the results to patients with SLE. Secondly, the relatively small size of the control and experimental groups of mice dictates the need to reproduce the results in a larger sample. Third, the increase (at weeks 8–22) and then decrease in cfDNA concentration in the control group (Figure 4a) and the delayed increase in anti-DNA Abs titers (Figure 4b) after the increase in cfDNA was revealed for the first time and needs to be verified in further studies.

## 5. Conclusions

This study revealed an inverse correlation of dynamic changes between total cfDNA and anti-dsDNA Abs in a pristane-induced mouse model of SLE. Thus, the levels of these markers are not correlated at individual time points, but their changes are inversely correlated, i.e., the more the anti-dsDNA Abs level increased, the more the cfDNA concentration decreased. These findings may indicate an important role of anti-dsDNA Abs in cfDNA clearance and the involvement of cfDNA in the pathogenesis of SLE. In addition, these data showed the dynamic nature of these indicators, so analysis at only one point may not be sufficiently informative. The direct correlation between dynamic changes in body weight and total cfDNA concentration in the plasma of pristane-immunized mice may indicate a negative role of excess adipose tissue formation in the pathogenesis of SLE.

## Figures and Tables

**Figure 1 pathophysiology-32-00048-f001:**
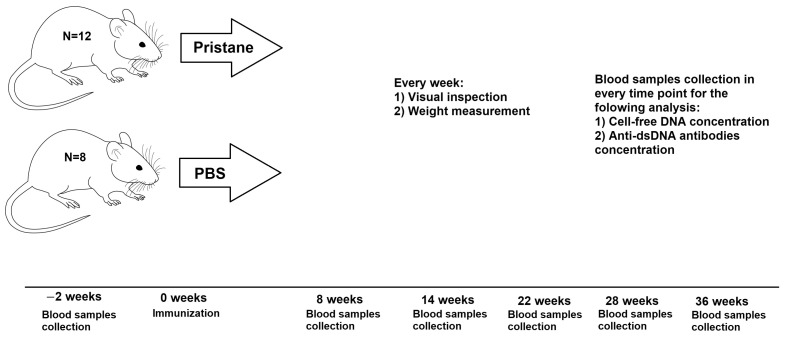
Experimental design.

**Figure 2 pathophysiology-32-00048-f002:**
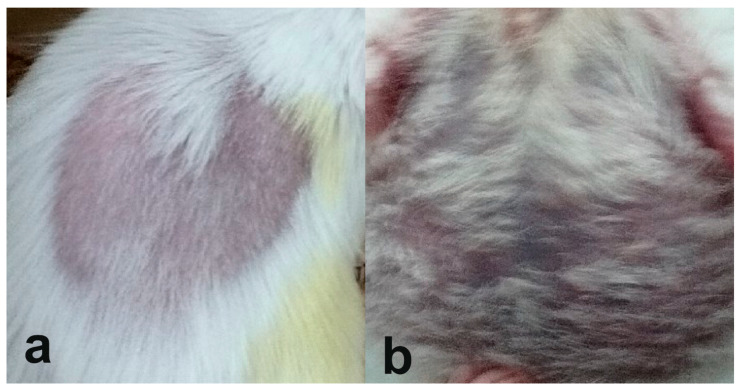
Changes in the appearance of mice associated with the development of SLE-like condition induced by immunization with pristane. Alopecia areata (**a**), loss of the hair coat, and accumulation of adipose tissue in the abdominal region (**b**).

**Figure 3 pathophysiology-32-00048-f003:**
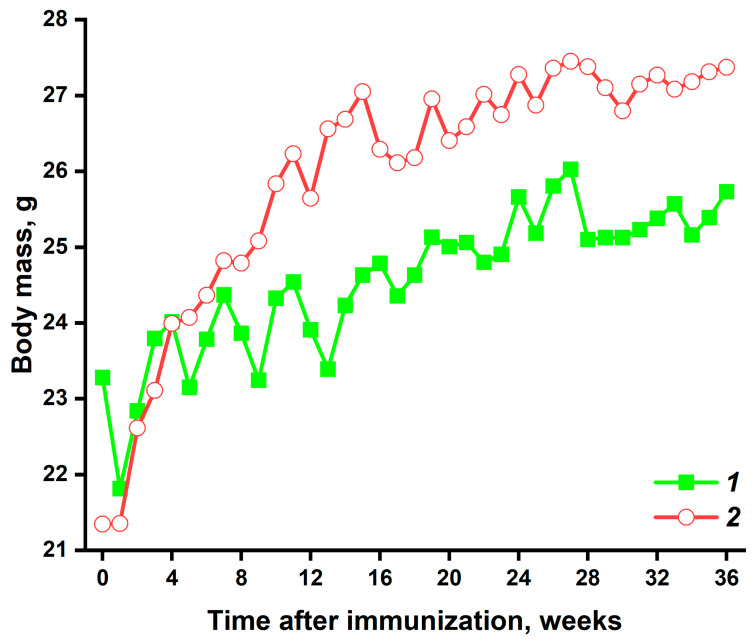
Weight change in experimental and control mice during the study. Points on the graph indicate median values. 1—PBS-immunized mice (*n* = 8), 2—pristane-immunized mice (*n* = 12). From week 8 (except at weeks 24 and 26), differences in weight were significant (not shown in the figure). The statistical significance of differences between the two groups at each time point was assessed using the Mann–Whitney test.

**Figure 4 pathophysiology-32-00048-f004:**
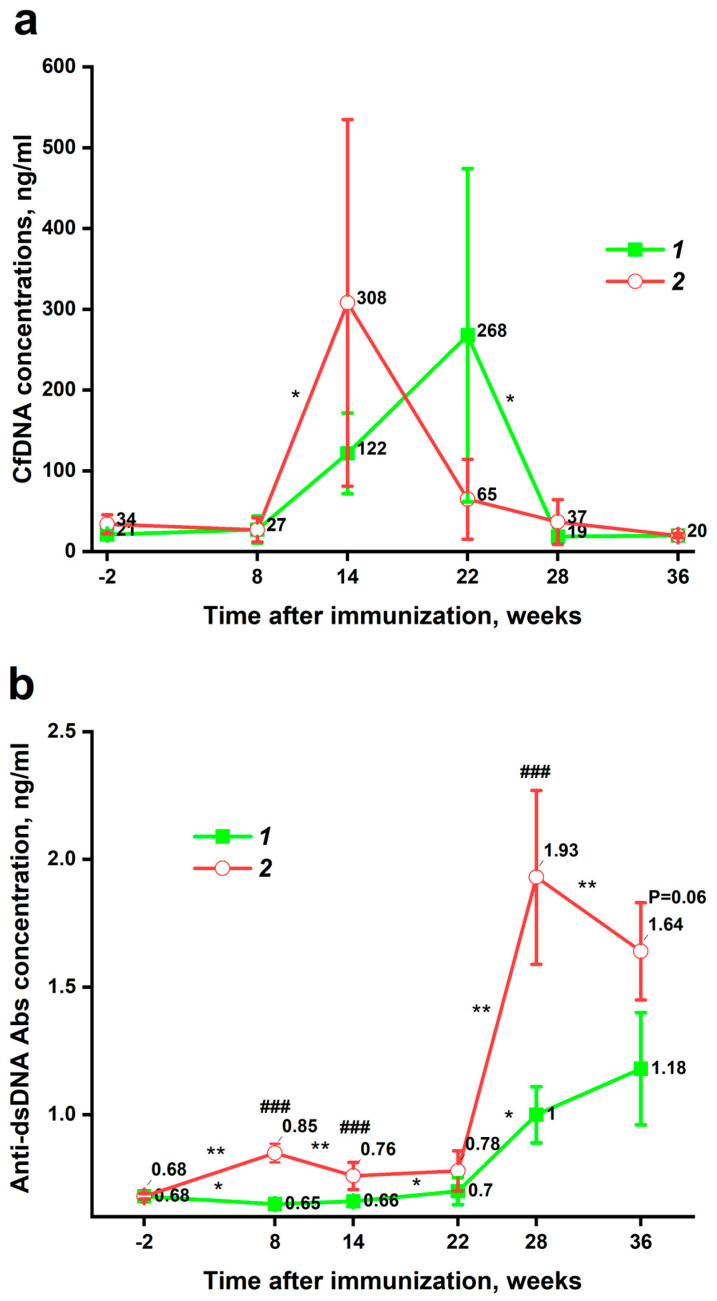
Plasma total cfDNA (**a**) and anti-dsDNA antibodies (**b**) concentration at six time points. Results are presented as median values. The whiskers on the graphs represent the absolute median deviation. 1—PBS-immunized mice (*n* = 8), 2—pristane-immunized mice (*n* = 12). The * symbol indicates a statistically significant difference between adjacent time points within the same group. Statistical significance was determined using the Wilcoxon test for paired samples. The # symbol indicates a statistically significant difference between the two groups at the same time point. Statistical significance of differences was assessed using the Mann–Whitney test. *—0.05 > *p* > 0.01; **—0.01 > *p* > 0.001; ###—*p* > 0.001.

**Figure 5 pathophysiology-32-00048-f005:**
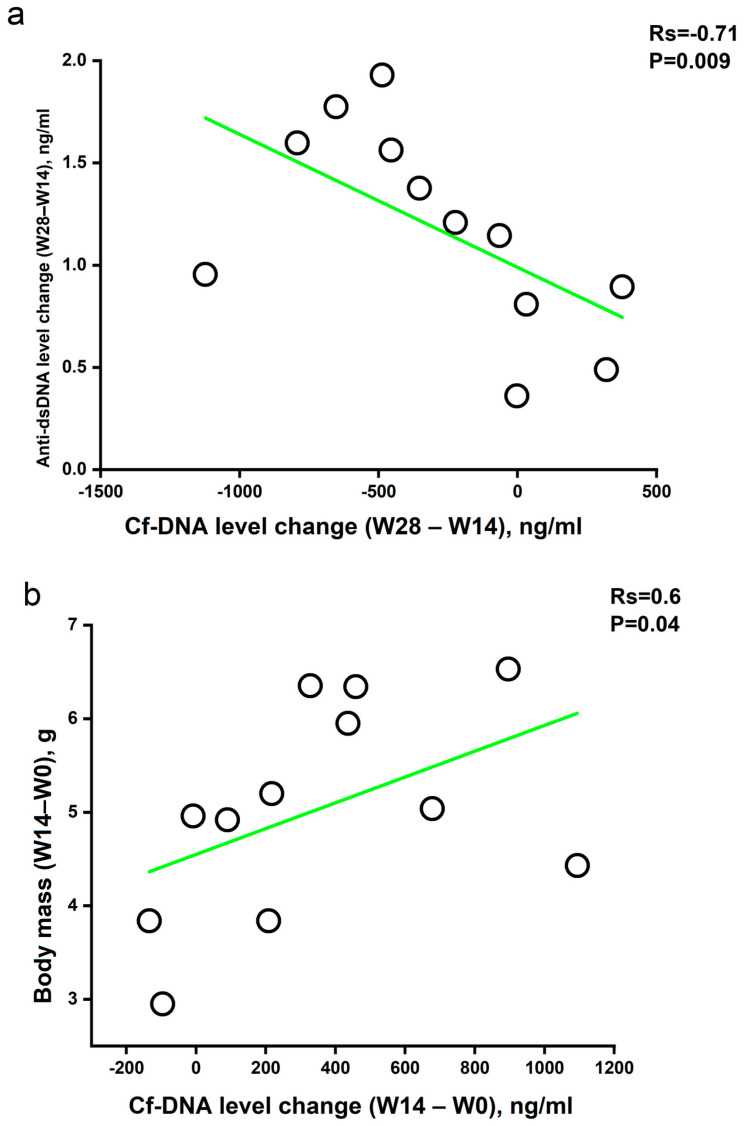
Correlation between the absolute change in the concentration (delta) of anti-dsDNA Abs and total cfDNA in plasma of mice from the experimental group (*n* = 12) from week 14 to week 28 after immunization (**a**). Correlation between the absolute change (delta) in body weight and cfDNA concentration in mice from the experimental group from week 0 to week 14 after immunization (**b**). Rs is Spearman’s correlation coefficient. Each dot represents changes in a specific mouse.

## Data Availability

Data supporting the reported results are presented in the manuscript. Additional and raw data are available on reasonable request from the corresponding author.

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
