# Peer review of "Dynamic Inverse Relationship Between Cell-Free DNA and Anti-dsDNA Antibodies in Experimental SLE Highlights the Potential for Targeted Immunomodulatory Therapy"

_pathophysiology, 2025, doi:10.3390/pathophysiology32030048_

Round 1
Reviewer 1 Report
Comments and Suggestions for Authors
The authors investigated the increase in the cell-free DNA level and its immune response by anti-DNA antibody formation and TNF-α secretion in a mouse SLE model for 38 weeks. They concluded that pristane-immunization caused high cfDNA levels, which decreased by anti-DNA abs and weight gain but did not affect the TNF-a secretion.
The article has a good flow with the introduction section, which describes the background, purpose, and necessity. The authors presented the study with complete details in sections material & methods and results. Finally, they discussed results based on the literature. Ultimately, the study contains information that needs to be complemented by future studies.
However, I should indicate a minor typo of “pristan-induced” in line 32.
Author Response
Dear Reviewer,
We thank the reviewer for the positive evaluation of our study.
Below we answer your comments point by point. Your comments are in italics.
The authors investigated the increase in the cell-free DNA level and its immune response by anti-DNA antibody formation and TNF-α secretion in a mouse SLE model for 38 weeks. They concluded that pristane-immunization caused high cfDNA levels, which decreased by anti-DNA abs and weight gain but did not affect the TNF-a secretion.
The article has a good flow with the introduction section, which describes the background, purpose, and necessity. The authors presented the study with complete details in sections material & methods and results. Finally, they discussed results based on the literature. Ultimately, the study contains information that needs to be complemented by future studies.
Reply: Thank you for your high appreciation of the manuscript. Indeed, our study provided the first results on the dynamic changes of cfDNA and anti-DNA antibodies in a pristane-induced mouse model of SLE. These results provide a starting point for further studies. For example, the effect of anti-DNA antibodies on the half-life of cfDNA could be studied, which would help to explain the identified patterns in this study.
However, I should indicate a minor typo of “pristan-induced” in line 32.
Reply: Thank you for noticing this typo. We have corrected this typo and have also done a thorough proofreading of the English.
Best regards
Authors
Reviewer 2 Report
Comments and Suggestions for Authors
This manuscript reports a study on the time-course relationship between plasma levels of anti-dsDNA antibodies (anti-dsDNA Abs) and cell-free DNA (cfDNA) in a pristane-induced mouse model of SLE. Also, the authors report measurements of TNF-α as an inflammatory mediator. The subject is highly relevant to the field of autoimmune diseases, and specifically to the understanding of the SLE pathogenesis where the roles of cfDNA and autoantibodies are not yet completely clear. The study is well designed and methodologically correct. The use of serial sampling over a 38-week period is another strength of the manuscript, as it allows for the study of temporal trends rather than cross-sectional snapshots. The finding of a negative dynamic correlation between changes in cfDNA and anti-dsDNA Ab levels, as opposed to correlation at single time points, is interesting and supports the hypothesis that cfDNA might be an antigenic stimulus that triggers the autoimmune response.
That being said, several aspects of the study need to be clarified and expanded in order to improve the scientific quality and clarity of the manuscript.
Firstly, the time course of the relationship between cfDNA and anti-dsDNA Abs is novel and well documented, but the biological explanation for the 14-week delay between the peak in cfDNA and the subsequent rise in anti-dsDNA Abs is not well developed. The authors state that this delay is longer than typical immune response times, but this needs to be elaborated. Is this delay due to the type of antigen presentation in this model, a slow accumulation of immune-stimulatory complexes, or possibly a limitation of the sampling intervals? Speculation on the mechanisms or references to similar lag phenomena in autoimmune models could improve the biological interpretation of this result.
A major concern is the fact that cfDNA levels also rose in the PBS-treated control group. This is an unexpected finding that questions the specificity of the pristane-induced cfDNA response and needs to be explored in more detail. It would be beneficial if the authors could explain whether environmental stressors, age-related changes, or non-pristane related immune activation could have caused this increase. Or was there any technical artefacts in the sample collection or cfDNA quantification that could explain this pattern?
The inclusion of TNF-α measurements seems redundant, as the data did not show any temporal or group-related changes. The purpose of measuring TNF-α is not clearly described in the introduction and is not well linked to the interpretation of the results. If TNF-α was thought to be a mediator of the cfDNA-induced immune response, a more coherent story should be given – either why TNF-α did not behave as expected or why it played such a minimal role in this study.
The data presentation is generally clear, but some suggestions for the improvement of the figures are made. All graphs should display statistical significance markers, and figure legends should include a description of variables, sample sizes and statistical tests used. Figures 4 and 6 are crucial for the study’s conclusions, and trend lines or annotations would enhance the comprehension of observed dynamics and correlations.
Another observation of note is the positive correlation between pristane-treated mice body weight and cfDNA levels. The manuscript proposes that increased adipose tissue may lead to increased inflammatory activity and cfDNA release. This is an intriguing possible link between metabolic and autoimmune processes, and the authors might consider expanding this discussion by referencing the literature on obesity-induced inflammation and cfDNA dynamics. However, interpretation should be guarded, as weight gain may be indicative of stress or non-specific immune activation rather than a lupus-specific process.
The statistical analysis seems appropriate, using nonparametric tests for small sample sizes. However, the small number of animals (n=12 experimental, n=8 control) creates concerns about the statistical power of the study. The authors should recognize this in the discussion and, if possible, do a power analysis to support the reliability of the correlations found.
The manuscript demonstrates good stylistic quality although it requires several grammatical edits and language improvements. The manuscript contains several terms which need correction such as “pristan-induced” instead of “pristane-induced.” The study's impact should be better represented through key numerical data points which include correlation coefficients and p-values in the abstract.
The research contributes to the expanding knowledge about extracellular DNA as an autoantibody production trigger which can benefit SLE disease monitoring and biomarker development. The experimental methodology demonstrates strength while the obtained results remain valuable. To maximize its potential the manuscript needs revisions with expanded discussion about control group cfDNA trends and the absence of TNF-α response together with the biological reasonability of the time delay observed in the study.
Author Response
Dear Reviewer,
The authors deeply appreciate your thorough analysis of our manuscript.
Below we answer your suggestions point by point. Your comments are in italics.
This manuscript reports a study on the time-course relationship between plasma levels of anti-dsDNA antibodies (anti-dsDNA Abs) and cell-free DNA (cfDNA) in a pristane-induced mouse model of SLE. Also, the authors report measurements of TNF-α as an inflammatory mediator. The subject is highly relevant to the field of autoimmune diseases, and specifically to the understanding of the SLE pathogenesis where the roles of cfDNA and autoantibodies are not yet completely clear. The study is well designed and methodologically correct. The use of serial sampling over a 38-week period is another strength of the manuscript, as it allows for the study of temporal trends rather than cross-sectional snapshots. The finding of a negative dynamic correlation between changes in cfDNA and anti-dsDNA Ab levels, as opposed to correlation at single time points, is interesting and supports the hypothesis that cfDNA might be an antigenic stimulus that triggers the autoimmune response.
Reply: Thank you for highlighting the key findings and strengths of our work.
That being said, several aspects of the study need to be clarified and expanded in order to improve the scientific quality and clarity of the manuscript.
Firstly, the time course of the relationship between cfDNA and anti-dsDNA Abs is novel and well documented, but the biological explanation for the 14-week delay between the peak in cfDNA and the subsequent rise in anti-dsDNA Abs is not well developed. The authors state that this delay is longer than typical immune response times, but this needs to be elaborated. Is this delay due to the type of antigen presentation in this model, a slow accumulation of immune-stimulatory complexes, or possibly a limitation of the sampling intervals? Speculation on the mechanisms or references to similar lag phenomena in autoimmune models could improve the biological interpretation of this result.
Reply: Thank you for this comment. We hypothesize that the 14-week delay is associated with slow accumulation of immune-stimulatory complexes, impaired T cell responses or involvement of T cell-independent antibody production mechanisms, and the activity of DNA clearance mechanisms including blood DNAases that fragment DNA reducing its antigenic potential. When such mechanisms are depleted, cfDNA becomes a more accessible antigen. The delay is possibly also due to differences in the type of antigen presentation compared to other SLE models. Finally, fluctuations of anti-dsDNA Abs may be related to a limitation of the sampling intervals. Unfortunately, we did not find similar lag phenomena in antibody formation in autoimmune models. We have added these considerations to the manuscript.
A major concern is the fact that cfDNA levels also rose in the PBS-treated control group. This is an unexpected finding that questions the specificity of the pristane-induced cfDNA response and needs to be explored in more detail. It would be beneficial if the authors could explain whether environmental stressors, age-related changes, or non-pristane related immune activation could have caused this increase. Or was there any technical artefacts in the sample collection or cfDNA quantification that could explain this pattern?
Reply: We believe that the increase in cfDNA levels in the PBS-treated control group cannot be attributed to artifacts in the sample collection or cfDNA quantification methodology. All cfDNA concentration studies were performed at one time point. Animals in the control group were kept under identical conditions. The influence of any external factors was excluded.
One possible reason for the increase in cfDNA concentration could be stress [30, references are given in the manuscript]. Another potential explanation for the rise in cfDNA levels is age. Human studies have shown that older individuals have higher cfDNA concentrations than younger ones [31]. For mice, this hypothesis is indirectly supported by the observation that anti-dsDNA abs levels increase with age. Thus, stress, age, and other factors can lead to fluctuations in cfDNA levels. It's also worth mentioning that unlike the experimental group, no statistically significant inverse correlation was found in the control group between the change in the concentration of cfDNA and anti-dsDNA Abs in the plasma of mice.
We have added these considerations to the manuscript (please see the third paragraph of the Discussion section).
The inclusion of TNF-α measurements seems redundant, as the data did not show any temporal or group-related changes. The purpose of measuring TNF-α is not clearly described in the introduction and is not well linked to the interpretation of the results. If TNF-α was thought to be a mediator of the cfDNA-induced immune response, a more coherent story should be given – either why TNF-α did not behave as expected or why it played such a minimal role in this study.
Reply: We agree that the inclusion of data on changes in TNF-α concentration seems redundant especially in light of the lack of significant differences. Therefore, following the recommendation of the third reviewer, we removed the results on TNF-α from the manuscript.
The data presentation is generally clear, but some suggestions for the improvement of the figures are made. All graphs should display statistical significance markers, and figure legends should include a description of variables, sample sizes and statistical tests used. Figures 4 and 6 are crucial for the study’s conclusions, and trend lines or annotations would enhance the comprehension of observed dynamics and correlations.
Reply: Thank you for this comment. Indeed, some figures were not accurate enough in terms of describing the statistics. Therefore, we have significantly revised Figures 3 and 4. In Fig 4 we indicated the significance of differences both within one group and between groups. In the caption to each figure, we added a description of the study groups, indicating n - the number of animals in each group, the statistical criterion used, and the level of significance.
Another observation of note is the positive correlation between pristane-treated mice body weight and cfDNA levels. The manuscript proposes that increased adipose tissue may lead to increased inflammatory activity and cfDNA release. This is an intriguing possible link between metabolic and autoimmune processes, and the authors might consider expanding this discussion by referencing the literature on obesity-induced inflammation and cfDNA dynamics. However, interpretation should be guarded, as weight gain may be indicative of stress or non-specific immune activation rather than a lupus-specific process.
Reply: Thank you for this interesting comment. The direct correlation between weight change and change in cfDNA concentration found in this study may be associated with the mechanism of neutrophil death known as netosis. Neutrophils are an important source of cfDNA released during netosis as part of neutrophil extracellular traps. Numerous studies indicate activation of netosis, including through adipokines and other mediators, in obese patients [presented in the manuscript]. Therefore, an increased netosis caused by obesity may be one of the reasons for the correlation found.
However, we agree that weight gain may be an indicator of stress or nonspecific immune activation.
We have added these considerations to the manuscript (please see penultimate paragraph of the Discussion section).
The statistical analysis seems appropriate, using nonparametric tests for small sample sizes. However, the small number of animals (n=12 experimental, n=8 control) creates concerns about the statistical power of the study. The authors should recognize this in the discussion and, if possible, do a power analysis to support the reliability of the correlations found.
Reply: Indeed, this study uses rather small samples of animals. This is because all guidelines for research involving animals recommend the use of a minimum number of animals. To determine the required sample size, a power analysis was performed. Since no literature data on cfDNA concentrations in pristane-immunized mice were found, human data from SLE patients were used as reference. In most studies of cfDNA concentration in patients with SLE, it was approximately 4-fold (or even more) higher than in healthy donors [9,22–24, references are cited in the manuscript]. Based on published studies, it was assumed that diseased animals would show a 4-fold increase in biomarker levels compared to controls. The standard deviation was estimated as 50% of the mean value in each group, corresponding to a standardized effect size of Cohen's d ≈ 1.41. With 80% power and a two-tailed significance level of α = 0.05, the calculation indicated that 7-9 animals per group would be required.
We've added information about power analysis to the manuscript (please see Lines 100-110 or 110-120 in the version with the specified edits).
We also indicated the small sample size as a limitation of this study.
The manuscript demonstrates good stylistic quality although it requires several grammatical edits and language improvements. The manuscript contains several terms which need correction such as “pristan-induced” instead of “pristane-induced.” The study's impact should be better represented through key numerical data points which include correlation coefficients and p-values in the abstract.
Reply: We have made careful proofreading of the English. Regarding the replacement of the term “pristane-induced” with “pristan-induced”, we assume you mistyped and meant the other way around. The generally accepted term is “pristane-induced”. The term “pristan-induced” does not appear in articles indexed in PubMed, while the term “pristane-induced” has 479 results. So we think you meant the term “pristane-induced”. The term “pristan-induced” was actually found once in the manuscript and we have replaced it with “pristane-induced”. Thank you for noticing that typo. Numerical data including correlation coefficients and p-values were also presented in Abstract.
The research contributes to the expanding knowledge about extracellular DNA as an autoantibody production trigger which can benefit SLE disease monitoring and biomarker development. The experimental methodology demonstrates strength while the obtained results remain valuable. To maximize its potential the manuscript needs revisions with expanded discussion about control group cfDNA trends and the absence of TNF-α response together with the biological reasonability of the time delay observed in the study.
Reply: Thank you for highlighting a number of unexpected patterns in the manuscript. To better explain these findings, we have expanded the discussion section (please see second paragraph of the Discussion section). We also revised the description of the limitations of this study.
Best regards
Authors
Reviewer 3 Report
Comments and Suggestions for Authors
Introduction:
- The rationale for choosing the pristane model over genetic models (e.g., MRL/lpr mice) is unclear.Briefly justify the pristane model’s relevance to human SLE (e.g., mimics autoantibody production, avoids confounding genetic factors).
- TNF-α’s role is underemphasized despite being measured. Add a sentence linking TNF-α to B-cell activation or cfDNA release to align with the study’s aims.
Methods:
- There are missing details on randomization and humane endpoints. So, specify how mice were allocated to groups (randomization method). Describe monitoring for distress (e.g., weight loss thresholds, euthanasia criteria).
- Blood collection volume per time point is unspecified in the manuscript. The authors have to write the volume drawn to assess impact on animal health.
Results:
- The cfDNA peak in controls (weeks 8–22) is unexplained. Please, discuss potential causes (e.g., aging, stress from repeated procedures) or label as a limitation.
- TNF-α results are negative but not contextualized. Therefore, address why TNF-α may not correlate with cfDNA/Abs (e.g., kinetic differences, compensatory pathways).
Discussion:
- The inverse cfDNA/anti-dsDNA correlation is highlighted, but mechanisms are speculative. The authors can propose testable hypotheses (e.g., antibody-mediated cfDNA clearance, cfDNA half-life modulation).
- The weight gain/cfDNA link lacks mechanistic insight. I suggest citing literature on obesity-related inflammation (e.g., adipokine effects on neutrophil extracellular traps).
The manuscript is well-written and scientifically clear, but there are minor grammatical, stylistic, and syntactical issues that could be polished for better readability and professionalism.
Author Response
Dear Reviewer,
The authors deeply appreciate your thorough analysis of our manuscript.
Below we answer your suggestions point by point. Your comments are in italics.
Introduction:
The rationale for choosing the pristane model over genetic models (e.g., MRL/lpr mice) is unclear.Briefly justify the pristane model’s relevance to human SLE (e.g., mimics autoantibody production, avoids confounding genetic factors).
Reply: Thank you for this comment. In this research, the pristane-induced model of SLE was used. This is one of the most popular induced models of the disease which has been applied for many years in the study of SLE pathogenesis and treatment. The pristane-induced lupus model is often contrasted with genetic models such as MRL/lpr mice. The pristine-induced model offers several key advantages [21, references are given in the manuscript]: it is easy to establish and maintain in a laboratory setting; pristane reliably induces lupus-like manifestations, including anti-dsDNA Abs production and kidney damage; immunized mice develop disease in a predictable timeframe (whereas the time between immunization and manifestation varies significantly in genetic models of SLE [21]), which is crucial for experimental planning; the development of lupus-like symptoms in this model is highly reproducible. The manifestation of lupus-like symptoms, especially anti-dsDNA Abs at predictable time frames with high reproducibility make this model particularly valuable when working with small cohorts of animals. These features make the pristane-induced model particularly valuable for mechanistic and therapeutic investigations in systemic lupus erythematosus
We have added these considerations to the manuscript (please see penultimate paragraph of the Intoduction section).
TNF-α’s role is underemphasized despite being measured. Add a sentence linking TNF-α to B-cell activation or cfDNA release to align with the study’s aims.
Reply: We believe that the inclusion of data on changes in TNF-α concentration seems redundant especially in light of the lack of significant associations. Therefore, we removed the results on TNF-α from the manuscript.
Methods:
There are missing details on randomization and humane endpoints. So, specify how mice were allocated to groups (randomization method). Describe monitoring for distress (e.g., weight loss thresholds, euthanasia criteria).
Reply: The mice were randomly divided into two groups: experimental and control. We used blinding to reduce the risk of bias in randomization. This was achieved by allocating animals to groups by a researcher unrelated to the experiment.
Rapid weight loss of more than 20% during the observation week was considered a humane endpoint requiring euthanasia. However, no animal met this criterion during the experiment.
We have added this data to the manuscript (please see section 2.1).
Blood collection volume per time point is unspecified in the manuscript. The authors have to write the volume drawn to assess impact on animal health.
Reply: Thank you for noticing that. It's a really important research aspect. Blood volume of 200 μl was collected through the retroorbital sinus. This volume did not exceed 10% of the total blood volume of the mouse. This volume of blood does not affect the animal's condition.
All manipulations were performed by an experienced researcher to minimize stress of the animal. Lidocaine was used as a topical ophthalmic anesthetic agent. All works with animals were performed in accordance with the recommendations of the Local Ethics Committee of the Institute of Chemical Biology and Fundamental Medicine, which are based on international recommendations.
Results:
The cfDNA peak in controls (weeks 8–22) is unexplained. Please, discuss potential causes (e.g., aging, stress from repeated procedures) or label as a limitation.
Reply: We believe that the increase in cfDNA levels in the PBS-treated control group cannot be attributed to artifacts in the sample collection or cfDNA quantification methodology. All cfDNA concentration studies were performed at one-time point. Animals in the control group were kept under identical conditions. The influence of any external factors was excluded.
One possible reason for the increase in cfDNA concentration could be stress [30, references are given in the manuscript]. Another potential explanation for the rise in cfDNA levels is age. Human studies have shown that older individuals have higher cfDNA concentrations than younger ones [31]. For mice, this hypothesis is indirectly supported by the observation that anti-dsDNA abs levels increase with age. Thus, stress, age, and other factors can lead to fluctuations in cfDNA levels. It's also worth mentioning that unlike the experimental group, no statistically significant inverse correlation was found in the control group between the change in the concentration of cfDNA and anti-dsDNA Abs in the plasma of mice.
We have added these considerations to the manuscript (please see the third paragraph of the Discussion section). We've also added this to the limitations.
TNF-α results are negative but not contextualized. Therefore, address why TNF-α may not correlate with cfDNA/Abs (e.g., kinetic differences, compensatory pathways).
Reply: As indicated above, the results for TNF-α were deleted.
Discussion:
The inverse cfDNA/anti-dsDNA correlation is highlighted, but mechanisms are speculative. The authors can propose testable hypotheses (e.g., antibody-mediated cfDNA clearance, cfDNA half-life modulation).
Reply: Thank you for this idea. We agree that the inverse cfDNA/anti-dsDNA Abs correlation may be related to antibody-mediated cfDNA clearance and thereby reduced cfDNA half-life.
We have added these considerations to the manuscript (please see the second paragraph of the Discussion section).
The weight gain/cfDNA link lacks mechanistic insight. I suggest citing literature on obesity-related inflammation (e.g., adipokine effects on neutrophil extracellular traps).
Reply: Thank you for this insightful comment. We agree that the direct correlation between weight change and change in cfDNA concentration found in this study may be associated with the mechanism of neutrophil death known as netosis. Neutrophils are an important source of cfDNA released during netosis as part of neutrophil extracellular traps. Numerous studies indicate activation of netosis, including through adipokines and other mediators, in obese patients [presented in the manuscript]. Therefore, an increased netosis caused by obesity may be one of the reasons for the correlation found.
We have added these considerations to the manuscript (please see penultimate paragraph of the Discussion section).
Comments on the Quality of English Language
The manuscript is well-written and scientifically clear, but there are minor grammatical, stylistic, and syntactical issues that could be polished for better readability and professionalism.
Reply: We have made careful proofreading of the English.
Best regards
Authors
Round 2
Reviewer 2 Report
Comments and Suggestions for Authors
I think the authors made sufficient changes in the manuscript. I believe the manuscript should be accepted.